# Comparing Machine Learning Classifiers for Predicting Hospital Readmission of Heart Failure Patients in Rwanda

**DOI:** 10.3390/jpm13091393

**Published:** 2023-09-18

**Authors:** Theogene Rizinde, Innocent Ngaruye, Nathan D. Cahill

**Affiliations:** 1College of Business and Economics, University of Rwanda, Kigali 4285, Rwanda; 2College of Science and Technology, University of Rwanda, Kigali 4285, Rwanda; ingaruye@gmail.com; 3School of Mathematics and Statistics, Rochester Institute of Technology, Rochester, NY 14623, USA; nathan.cahill@rit.edu

**Keywords:** HF, hospital readmission, ML algorithm, Rwanda

## Abstract

High rates of hospital readmission and the cost of treating heart failure (HF) are significant public health issues globally and in Rwanda. Using machine learning (ML) to predict which patients are at high risk for HF hospital readmission 20 days after their discharge has the potential to improve HF management by enabling early interventions and individualized treatment approaches. In this paper, we compared six different ML models for this task, including multi-layer perceptron (MLP), K-nearest neighbors (KNN), logistic regression (LR), decision trees (DT), random forests (RF), and support vector machines (SVM) with both linear and radial basis kernels. The outputs of the classifiers are compared using performance metrics including the area under the receiver operating characteristic curve (AUC), sensitivity, and specificity. We found that RF outperforms all the remaining models with an AUC of 94% while SVM, MLP, and KNN all yield 88% AUC. In contrast, DT performs poorly, with an AUC value of 57%. Hence, hospitals in Rwanda can benefit from using the RF classifier to determine which HF patients are at high risk of hospital readmission.

## 1. Introduction

Heart failure (HF) occurs when the heart becomes too weak or stiff to effectively pump blood to meet the body’s needs [1], resulting in a variety of health problems and high medical expenses. HF has a significant global impact, affecting millions of people worldwide despite medical advancements [2,3]. There is a shortage of HF data on certain patient populations [3]: Many studies focus on HF patients in the United States, but these studies might not be representative of HF patients in the rest of the world [4,5,6]. In Africa, HF is still a significant clinical and health concern, often manifesting as an urgent medical condition requiring prolonged hospital stays [7]. Comprehensive data on HF are lacking in Sub-Saharan Africa (SSA), with little information primarily obtained from urban hospitals [8,9,10]. SSA HF cases have high hospitalization rates and significant associated healthcare costs. Like other countries, HF is a serious public health emergency [11] in Rwanda. Non-communicable illnesses such as HF accounted for 34.7% of deaths in Rwanda in 2020. The need to address HF’s socio-economic effects is highlighted by the fact that it accounts for 5% to 10% of adult hospital admissions in SSA, with a similar trend in Rwanda [12].

Several methods for lowering HF-related hospital readmissions have been explored over the past three decades [13,14]. However, there is still a lack of widespread application machine learning (ML)/artificial intelligence (AI) techniques for anticipating readmissions due to HF, particularly in low- and middle-income countries [15]. ML classifiers enable computers to autonomously learn from data, recognize patterns, and predict outcomes from various inputs without explicitly being programmed [14]. The popularity of ML has grown across a variety of industries due to its outstanding ability to quickly analyze large datasets and reach complex conclusions, improving operations, data-driven decision-making, and innovation [16].

ML classifiers have been used in previous research to precisely predict outcomes like hospital readmission for HF and others [17,18]. However, issues with limited electronic health data integration and class imbalance in medical datasets are present in low- and middle-income countries [19]. Due to lack of data unique to specific countries, like health knowledge, cultural norms, and medical facilities, Rwanda lacks reliable and accurate predictive models to be used in clinical practice [17,20,21]. In this study, we use locally gathered comprehensive data on HF in Rwanda, and we explore a variety of ML classifiers to predict HF hospital readmission. We compare prediction performance of multi-layer perceptrons (MLP), logistic regression (LR), decision trees (DT), K-nearest neighbors (KNN), random forests (RF), and support vector machines (SVM). This study also seeks to pinpoint crucial factors influencing hospital readmissions for HF in Rwanda, providing knowledge and skills to enhance the management of HF. Efforts to accurately predict HF hospital readmission, and to identify high-risk HF patients in Rwanda, may improve HF management and result in better patient outcomes and cost savings [22,23].

## 2. Materials and Methods

This retrospective study collected data from medical records of HF patients who were hospitalized in Rwanda between 1 January 2008 and 31 December 2019. The records were obtained from seven hospitals that were able to treat HF in Rwanda. These include Rwandan Military Hospital (RMH), King Faisal Hospital (KFH), University Teaching Hospital of Butare (CHUB), University Teaching Hospital of Kigali (CHUK), Rwinkwavu Hospital (RWH), Kirehe Hospital (KIH), and Butaro Hospital (BUH). We extracted various features of interest from the patients’ medical records, including age, sex, district of residence, marital status, occupation, resting heart rate, blood pressure, history of hypertension and smoking, heart ultrasound results, risk factors for HF, number of hospitalization days, respiratory rate upon admission, slope, chest pain, cholesterol status, blood sugar, results of electrocardiography at rest, reason for discharge, presence of feces on admission, and past medical and family history.

We utilized Jupyter Notebook as the primary tool for building ML models. We installed Python 3.11.1 and essential packages such as pandas, seaborn, matplotlib, and scikit-learn, which come with built-in libraries and functions, to facilitate data manipulation, visualization, analysis, and construction of machine learning models. Jupyter Notebook was our preferred tool due to its user-friendly interface, advanced data cleaning capabilities, and fast implementation of modeling processes using the Python programming language.

In this study, we compared six ML classifiers, including multi-layer perceptron (MLP), K-nearest neighbors (KNN), logistic regression (LR), decision trees (DT), random forests (RF), and support vector machines (SVM). All classifiers were trained, tested, and compared using performance metrics including the area under the receiver operating characteristic curve (AUC), sensitivity, and specificity. Inputs for each classifier included the following: The MLP classifier contained 3 hidden layers with 128, 64, and 32 neurons in the hidden layers, and it used a sigmoid activation function. The KNN classifier assumed k = 5; the DT classifier used a maximum depth of 8 and a minimum of 20 samples per leaf. For splitting purposes, Gini impurity criteria were used. The RF classifier used a random forest ensemble with 100 trees, with a maximum depth of 10 and 10 samples per leaf. Then, the SVM classifier with a radial basis function kernel used a regularization parameter of 10. For the SVM with a linear kernel, the regularization parameter was 0.1; other hyperparameters were considered at their default values for simplicity.

First, we performed data collection, exploratory data analysis, and preprocessing to clean and prepare the data for further analysis. Here, we dropped all variables with 50% or more null values from the data frame. For most variables with less than 50% null entries, we filled in missing values using the KNN imputer algorithm. For the age variable, we filled in the missing values using the median since this variable appeared to have a uniform distribution. Second, the imbalance in the dataset was handled to ensure that the two classes of HF patients (0 = no hospital readmission; 1 = at least one hospital readmission within 20 days of hospital discharge) were balanced. To address the problem of dataset heterogeneity, we used the Synthetic Minority Oversampling Technique (SMOTE). SMOTE is an oversampling method that involves synthesizing new instances using the current data to oversample underrepresented groups [24,25]. Third, we extracted important features to reduce the dimensionality of the dataset as it contained over 60 features. This step ensured that only relevant features were included in the model development process, leading to improved model accuracy. We then split the dataset into training and testing sets to evaluate the performance of the model. According to the methodology of Dobbin and colleagues, we used 80% of the data for training and 20% for testing [26,27]. Fourth, we standardized the features for training and testing by shifting and scaling the data to have zero mean and unit standard deviation. Lastly, we trained the classifiers with the training set and evaluated using the testing set to determine the precision and accuracy of the models in predicting HF hospital readmission rates in Rwanda.

We used a confusion matrix as the evaluation metric to measure the performance of the algorithms on both the training and testing datasets. Furthermore, the receiver operating characteristic curve (ROC), the area under the ROC curve (AUC), accuracy, precision, recall, and F1-score metrics were utilized to plot, compare, and identify the best-performing model. The results of this evaluation provided the necessary information for drawing conclusions in line with the research objectives.

Though there was no direct contact with the HF patients while collecting HF data but with their respective files, ethical approval was provided by competent authorities including the Ministry of Health of Rwanda and the Rwandan Institutional Review Board and all concerned seven hospitals.

## 3. Results

### 3.1. Exploratory Data Analysis and Preprocessing

We loaded the dataset into the Python Jupyter Notebook, a Python environment where we conducted exploratory data analysis, preprocessing, and model building. The data frame shown in Table 1 contains information for the top five records, including 75 features.

The next step was to conduct exploratory data analysis. Upon inspection, we discovered that almost all the features in the data frame were of string data type, and the dataset contained a total of 4085 objects. To facilitate further analysis, visualization, and preprocessing, we converted the variables to numeric types and subdivided the entries with class labels.

One aspect of our analysis focused on the distribution of admitted HF patients’ age as illustrated in Figure 1.

As shown by Figure 1, patients in their late fifties (50 s) to late sixties (60 s) had the highest admission rates, followed by patients between the ages of zero (0) and early twenties. The least-admitted patients due to HF were aged 90 years and above.

Figure 2 show that the collected and pre-processed dataset was imbalanced, with class 0 accounting for 85% of the total dataset with 3469 records.

Training a model on this imbalanced dataset without addressing the imbalance could result in accurate predictions because the model will perform poorly on class 1, leading to inaccurate overall predictions. Therefore, the imbalance was addressed using the SMOTE oversampling algorithm. The results of the target class before and after handling the imbalance are presented in Table 2.

### 3.2. Important Features Selection

To optimize the prediction performance of our ML models, we need to mitigate overfitting by selecting important or influencing features. Using the ExtraTreesClassifier tool in python, random splits of all the observations were performed to avoid undesirable ML behavior. This is a score for the preprocessed dataset, displaying only the ten most significant features out of the 59 preprocessed features.

By order of importance, crucial features identified in the Figure 3 include the district of residence (district), shortness of breath (symptom_1), maximum diastolic blood pressure at rest (dbp_max_resrtbpr), maximum systolic blood pressure at rest (sbp_max_rerstbpr), maximum heart rate (maxhra), risk factor of decompensated heart failure called cardiac arrthym (risk_decom_5) alcohol intake (alcohol), sex of the HF patient (sex), number of days for the first hospitalization(hosp_days), and age(age) of the HF patient. These features play a critical role in determining the likelihood of a patient with HF to be readmitted to the hospital.

### 3.3. Model Building and Evaluation

#### 3.3.1. Random Forest Classification

The confusion matrix shown by Table A1 in Appendix A indicates that the Randon Forest (RF) classification model achieved a 100% accuracy rate when predicting both classes 0 and 1 using training set. Additionally, the precision, recall, and f1-score metrics all achieved a score of 100%. The training set accuracy for this model is also 100%. On the other hand, while using the new data, i.e., the testing set, as shown by Table A2 in Appendix A, the confusion matrix reveals that the RF model correctly predicted 599 instances of class 0 but incorrectly predicted 109 instances of class 0.

For the new data or testing set, again the same Table A2 in Appendix A shows that the model incorrectly predicts 64 instances of class 1 and correctly predicts 616 instances of class 1. The precision of the model in class 0 is 89%, and in class 1, it is 84%. The recall score in class 0 is 84%, and in class 1, it is 89%. The f1-score is 0.87 in both class 0 and class 1. The accuracy of the RF model in the testing set is 87% as well.

Figure 4 depicts the AUC and ROC of the RF classifier. The figure shows that the AUC value is 0.94, and this implies that the model’s predictions are largely accurate and also that the probability of obtaining a wrong prediction from the model is 0.06. Although on a small scale, this wrong prediction is negligible; as the sample size increases, the wrong predictions are certain to take a toll on the model’s performance. The RF classifier has a low false-positive rate and a high true-positive rate if the AUC is high. This indicates that the RF classifier has a strong ability to categorize cases that are present (true positives) and that it also has a strong ability to classify situations that are absent (true negatives). This translates into the classifier having a high predictive accuracy, making it a useful tool for determining the pertinent features that go into the target variable.

#### 3.3.2. SVM Classification Using Linear Kernel

The performance of the SVM classifier using a linear kernel was evaluated, and Table A3 in Appendix A presents the resulting confusion matrix and classification report for the training data. The SVM model with a linear kernel made 2149 correct predictions and 612 incorrect predictions for class 0, and for class 1, there were 2346 correct predictions and 443 incorrect predictions. The overall accuracy of the model in the training data is 81%, with a precision of 83% for class 0 and 79% for class 1.

On the other hand, using the testing data as presented in Table A4 in Appendix A, the SVM model with a linear kernel made 540 correct predictions and 168 incorrect predictions for class 0. For class 1, the model made 558 correct predictions and 122 incorrect predictions. The overall accuracy of the model in the testing set is 79%.

Figure 5 reveals that the AUC value is 0.88, which suggests that the classifier’s predictions are reasonably accurate and that there is a 0.12 chance of generating a prediction that is incorrect. Once the size of the dataset quadruples, this incorrect prediction will undoubtedly have an adverse effect on the model’s performance. It is crucial to keep in mind, though, that an AUC score of 0.88 does not imply that the SVM model is definitely the best model for the specific situation. Simply said, it indicates that the model is successful at classifying data according to the desired variable. Depending on the particular situation, various model performance measures may be assessed. In order to choose a model with confidence, it is vital to thoroughly compare various models using various performance indicators.

#### 3.3.3. SVM Classification with Gaussian Radial Basis Function Kernel

Using the training data Table A5 in Appendix A shows the confusion matrix of the SVM model with a Gaussian radial basis function (RBF)kernel. It indicates that out of a total of 2761 predictions made, 2162 were valid, and 599 were false positives for class 0. Similarly, for class 1, the model made 2354 correct predictions and 435 false predictions. The overall prediction accuracy of the model is 81%.

The precision of the model in class 0 is 83%, and in class 1, it is 80%. On the other hand, using the testing data, Table A6 in Appendix A displays the confusion matrix and classification report for the SVM with Gaussian Radial Basis Function Kernal. The results in Table A6 in Appendix A indicate that the model correctly predicts 556 instances of class 0 but incorrectly predicts 152 instances of class 0. For class 1, the model correctly predicts 591 instances and made 89 false predictions. The overall accuracy of the model in the testing or new data is 83%. The precision of the model in class 0 is 86%, while in class 1, it is 80%.

Figure 6 demonstrates that the AUC value is 0.87, which suggests that the classifier’s predictions are reasonably accurate and that there is a 0.13 chance of generating a prediction that is incorrect.

Once the size of the dataset quadruples, this incorrect prediction will undoubtedly have an adverse effect on the model’s performance. It is crucial to keep in mind, though, that an AUC score of 0.88 does not imply that the SVM model is definitely the best model for the specific situation. Simply said, it indicates that the model is successful at classifying data according to the desired variable. Depending on the particular situation, various model performance measures may be assessed. In order to choose a classifier with confidence, it is vital to thoroughly compare various models using various performance indicators.

#### 3.3.4. Evaluation of the KNN Classifier

The K-nearest neighbors (KNN) model evaluation on the training data is as shown in Table A7 in Appendix A.

It displays the confusion matrix and classification report of the KNN model on the training dataset. The model made 2319 correct predictions and 442 incorrect predictions for class 0, and for class 1, there were 2680 correct predictions and 109 incorrect predictions. The overall accuracy of the model in the training dataset is 90%. On the other hand, Table A8 in Appendix A presents the confusion matrix and classification report of the KNN model on the testing dataset.

The model’s accuracy in the testing data is 85%, with 554 correct predictions and 154 incorrect predictions for class 0. For class 1, the model made 621 correct predictions and 59 incorrect predictions.

Figure 7 shows that the AUC value for the KNN classifier is 0.88, which is nearly identical to that of the SVM (with linear kernel), and in the same vein, it signifies that the model’s predictions are largely accurate and that there is a 0.12 probability of receiving an inaccurate forecast of HF hospital readmission. The KNN model’s AUC of 0.88 further indicates that the model can accurately and confidently distinguish between positive and negative instances.

#### 3.3.5. Evaluation of the DT Classifier

Table A9 in Appendix A, indicates that the DT classifier has a prediction accuracy of 54% in the training data.

The model performed poorly on class 0, as it incorrectly predicts almost the entire class 2503 and correctly predicts only 258 but performs relatively well on class 1 by correctly predicting almost the entire class of 2724 and incorrectly predicts only 65 instances. The precision, recall, and f1-score of the model on class 0 are 80%, 9%, and 17%, respectively. This indicates that the model did not learn much about class 0. The overall performance is 54%. Likewise, on the testing set, the model also performs poorly, as it did not learn well in either class 0 or class 1. Table A10 in Appendix A provides a detailed confusion matrix and classification report of the decision tree model on the testing set. The overall performance of this model on the testing data is 52%, which is the lowest in these trained models.

For the area under the ROC curve, Figure 8 shows an AUC of 0.57, which indicates that the DT classifier is not trustworthy enough to provide solid predictions of HF hospital readmission in Rwanda.

Simply said, the model is just slightly more accurate than speculating. It is not a reliable model for making critical judgments or predictions of HF hospital readmission in Rwanda. With an AUC of 0.57, the decision tree model cannot be trusted to make reliable predictions.

#### 3.3.6. Evaluation of the Logistic Regression (LR) Classifier

Table A11 in Appendix Ashows the evaluation of the LR classifier on the training part of the dataset. The model correctly predicts 2056 instances of class 0, but incorrectly predicts 705 instances of the same class. Again, the model correctly predicts 2171 instances of class 1, but incorrectly predicts 618 instances of class 1. The overall accuracy of the model in the training set is 76%, with a precision of 77% for class 0 and 75% for class 1, as well as a recall of 74% for class 0 and 78% for class 1.

The performance of the logistic regression model on the testing set is illustrated in Table A12 in Appendix A. The findings revealed that the model achieves an accuracy score of 77% in its predictions. In addition, the same table indicates that the model makes 531 correct predictions on class 0 and 177 incorrect predictions, while for class 1, the model makes 532 correct predictions and 148 incorrect predictions. The precision of the model in class 0 is 78%, and in class 1, it is 75%. Then again, Figure 9 shows a LR classifier with an AUC of 0.81, which indicates that the model has a decent ability to distinguish between the readmitted and non-readmitted HF patients.

An AUC score of 0.5 is similar to random guessing, whereas an AUC score of 1 denotes a perfect classifier. As a result, an AUC of 0.81 shows that the model is reasonably effective at predicting the response variable and is better than random guessing. Here, the positive class denotes those who are prone to HF readmission, whereas the negative class denotes those who are not. An AUC of 0.81 means that it can accurately classify 81% of patients as subject to readmission; in other words, the model’s error rate of 19% indicates that it predicts both positive and negative classes with the same level of error.

#### 3.3.7. Model Building Using Multilayer Perceptron Model

Table A13 in Appendix A presents the evaluation of the multilayer perceptron (MLP) model on the training set. The model correctly predicted 2476 instances of class 0 but incorrectly predicted 285 instances of class 0 as class 1. The model also correctly predicted 2615 instances of class 1 but incorrectly predicted 174 instances of class 1 as class 0. The overall accuracy of the model in the training set is 92%, with a precision of 93% for class 0 and 90% for class 1, as well as a recall of 90% for class 0 and 94% for class 1.

Furthermore, the performance of the MLP classifier on the testing set is illustrated in Table A14 in Appendix A. The model achieved an accuracy score of 82% in its predictions. The confusion matrix indicates that the model made 552 correct predictions on class 0 and 156 incorrect predictions, while for class 1, the model made 583 correct predictions and 97 incorrect predictions. The precision of the model in class 0 is 85%, and in class 1, it is 79%.

Figure 10 displays the MLP’s AUC and ROC. It illustrates the model’s AUC value of 0.88, which shows that predictions are only marginally accurate and that there is a 0.12 probability of receiving an inaccurate one.

An AUC of 0.88 produced by the MLP model is regarded as a reasonably good result, showing that the model can be used practically and has a respectable ability to distinguish between the positive and negative classes. However, by tweaking the model or including more elements in the data, more advancements might be achievable.

### 3.4. General Result Evaluation and Comparison

Each algorithm utilized to predict hospital readmission rates for HF in Rwanda has its own unique characteristics and tendencies. The review process aims to narrow down the choices to the most promising ones that can be implemented in Rwanda. During the evaluation process, the accuracy score and overall performance on both the training and testing datasets are considered when comparing the algorithms. Thus, the study concludes with evidence in the form of different results obtained from assessing all the models.

Figure 11 provides a comparison of the performance of all the models using the receiver operating characteristics (ROC) curve and the area under the curve (AUC) metric. From the graph, it is evident that the random forest model outperforms the other models, with an AUC score of 0.94. The KNN and SVM models follow with an AUC score of 0.88, while the logistic regression and decision tree models exhibit the worst performance, with AUC scores of 0.81 and 0.57, respectively.

The results in the Table 3 show that the area under the ROC curve for the decision trees model is 57%, which is the worst performance compared to the seven trained models. On the other hand, the random forest model outperformed all the trained models, with the area under the ROC curve of 94%. It is very important to mention that all four classifiers (SVM with linear kernel, SVM with Gaussian radial basis function kernel, KNN, and MLP) ranked second, with an excellent performance, having the area under the ROC curve of 88%; the LR classifier ranked third, with the area under the ROC curve of 81%.

The findings in the Figure 12 demonstrate that the choice of the model significantly affects how well the classification tasks are carried out. Therefore, it is very important to test various models and contrast the results to be able to identify the most accurate model that satisfies the dataset’s specific requirements.

Table 3, Figure 11 demonstrate that random forest outperforms other models, with an AUC of 94% and an evaluation accuracy of 87%. KNN and SVM models also show excellent performance, both having an AUC of 88%, followed by logistic regression, with an AUC of 84%. On the other hand, the decision tree model has the worst performance, with an AUC of 57%, similar to the baseline model.

## 4. Discussion

The purpose of this study was to investigate the applicability of multiple ML classifiers for predicting HF hospital readmission in Rwanda using locally gathered data. Our findings revealed notable insights into the effectiveness of these models. In terms of predicting hospital readmissions in the Rwandan context among the models evaluated, the random forest classifier emerged as the most promising option for predicting HF hospital readmission in Rwanda. In addition, the support vector machine, K-nearest neighbors, and multi-layer perceptron approaches all demonstrated admirable performance. Therefore, the results obtained in this study are consistent and reliable, which is supported by the fact that they align well with the study conducted in 2022 by Michailidis and colleagues [28]. This implies that the effectiveness of these classifiers can be, at least in part, generalized and is not limited to a particular dataset or context. On the other hand, it is crucial to note that the decision tree classifier only managed to achieve an area under the curve of 57%, and this performance is below average. This outcome differs significantly from the performance of the same model that has been used previously in the literature [29,30,31]. This might indicate that the decision tree classifier is not a good fit for the specific characteristics of the healthcare dataset from Rwanda.

Generally, the findings of this study as a whole highlight the potential of ML techniques in accurately predicting hospital readmission for HF patients in Rwanda. Nevertheless, system-specific challenges will need to be carefully considered in future studies due to the decision tree classifier’s poor performance, which calls into question its suitability. In fact, the use of these models can lead to better long-term health outcomes, reduced readmissions to hospitals, and enhanced patient care. These predictive capabilities might help healthcare professionals in Rwanda allocate resources more effectively and customize interventions to patients’ unique needs, thereby improving the standard of care. Nevertheless, it is important to recognize the limitation and challenges that come up when applying ML techniques to the Rwandan healthcare system. These classifiers’ performance and generalizability may be impacted by the particularities and complexities of the Rwandan healthcare system, including data availability, quality, and cultural considerations.

## 5. Conclusions and Recommendations

In order to improve the standard of care and health outcomes of HF in Rwanda, more research is still required. This can result in more accurate predictions, personalized treatment plans, and better utilization of healthcare resources. Future research could improve classifiers to better fit the local context, address data issues, and account for the unique constraints of the healthcare system. Addressing the identified knowledge gaps can aid in the development of more precise and relevant predictive models for HF readmission in Rwanda, which can have a positive impact on the country’s ability to manage and prevent HF hospital readmission.

While random forest classification shows the best performance, it is important to base actions on principles. The SVM approach also works well for this purpose. The study’s results indicate that ML techniques can accurately predict hospital readmission for HF patients in Rwanda, which can lead to improved care, fewer hospital readmissions, and better long-term health outcomes.

In order to help healthcare practitioners anticipate the possibility of readmission for specific patients, the classifiers can be used as decision support tools. By offering early interventions and personalized treatment plans, this technology can assist healthcare professionals in making the most use of their resources and improving patient outcomes. RF is a useful tool for anticipating HF readmissions, offering a strategy that can help patients obtain better results and make the most of healthcare resources. To minimize potential negative effects, such as over-reliance on the model’s predictions, which would impair clinical judgement and patient-centered treatment, healthcare practitioners must understand the model’s strengths and limits and ensure that it is used effectively.

To conclude, this study contributes to the growing body of literature on the application of the ML algorithm in medicine and suggests that ML has the potential to enhance HF management in Rwanda.

## Figures and Tables

**Figure 1 jpm-13-01393-f001:**
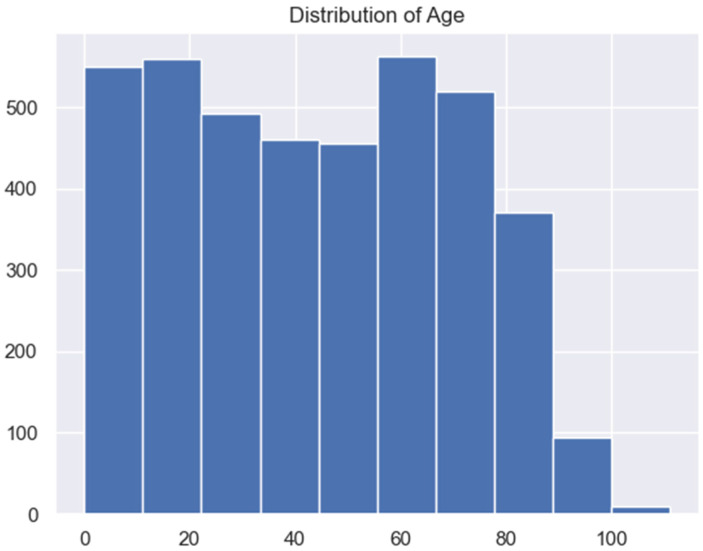
Admitted patients per age.

**Figure 2 jpm-13-01393-f002:**
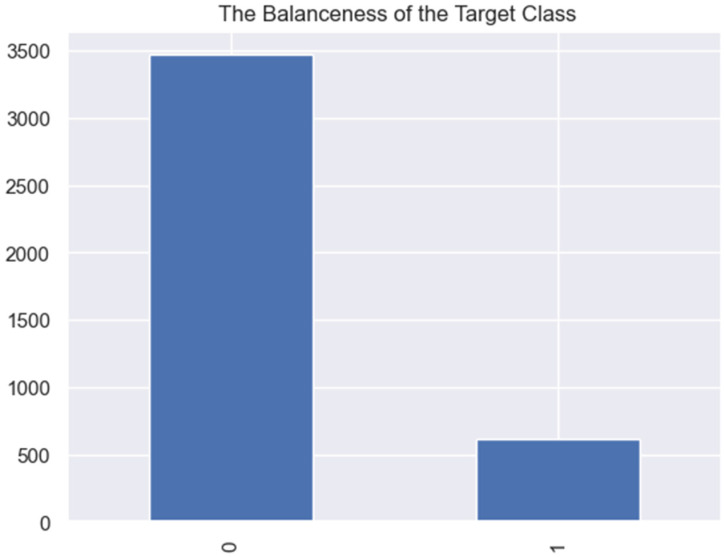
Imbalances in the target class before using SMOTE in handling the imbalance.

**Figure 3 jpm-13-01393-f003:**
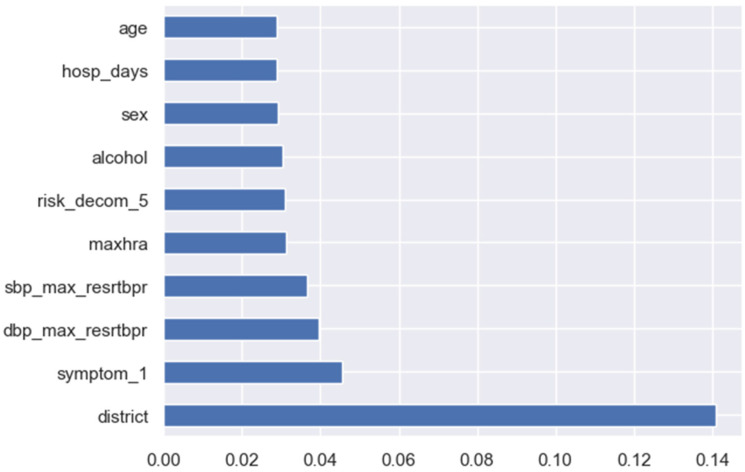
Graph of features and their importance.

**Figure 4 jpm-13-01393-f004:**
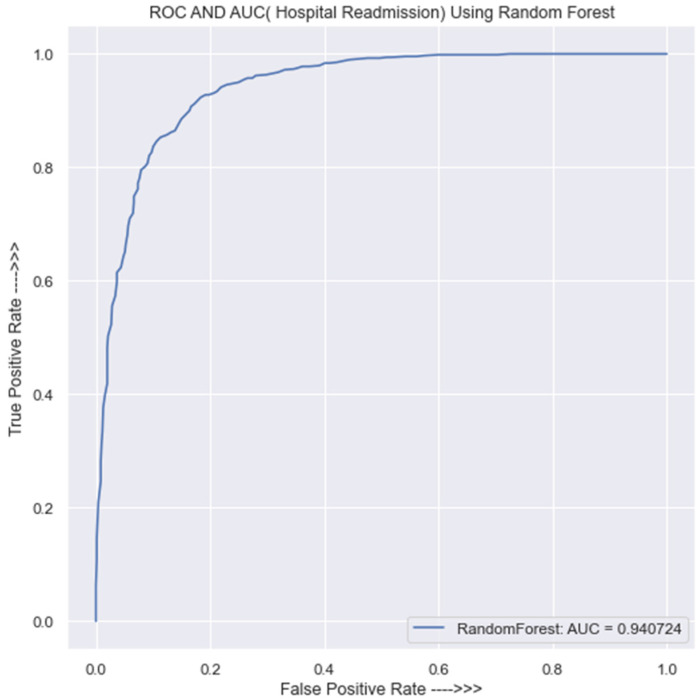
ROC curve for RF classifier.

**Figure 5 jpm-13-01393-f005:**
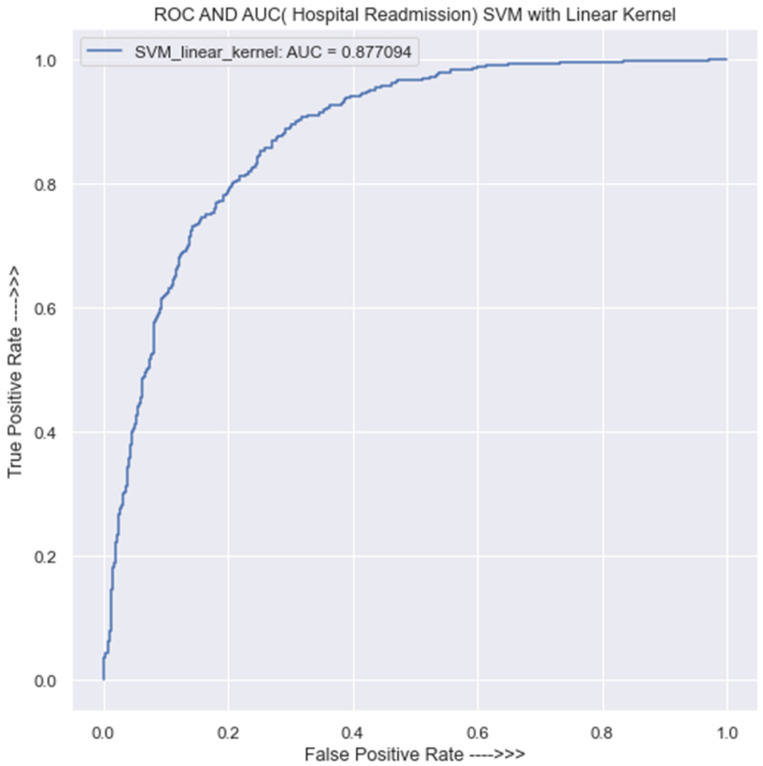
ROC curve for SVM classification using the linear kernel.

**Figure 6 jpm-13-01393-f006:**
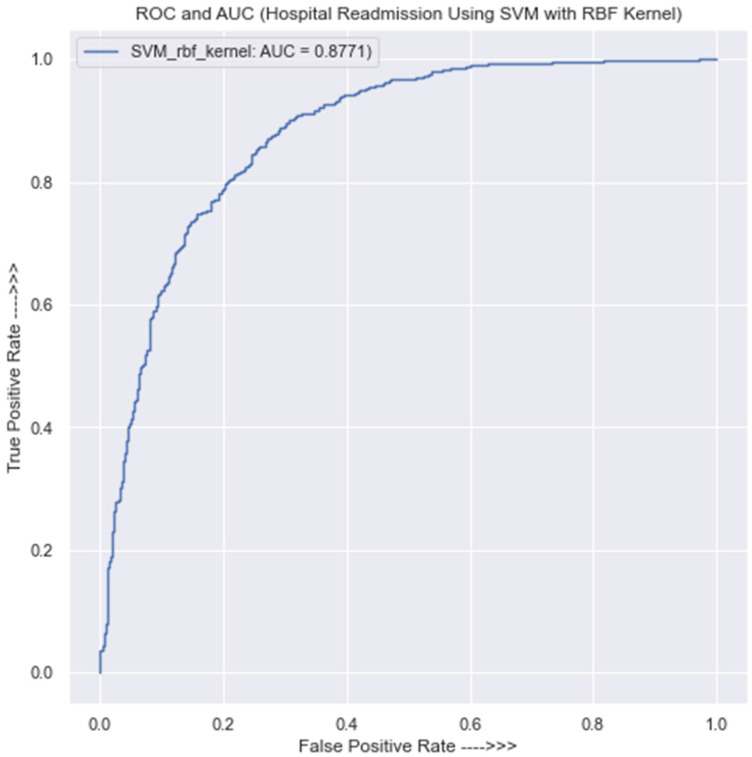
ROC curve for SVM classifier with Gaussian radial basis function kernel.

**Figure 7 jpm-13-01393-f007:**
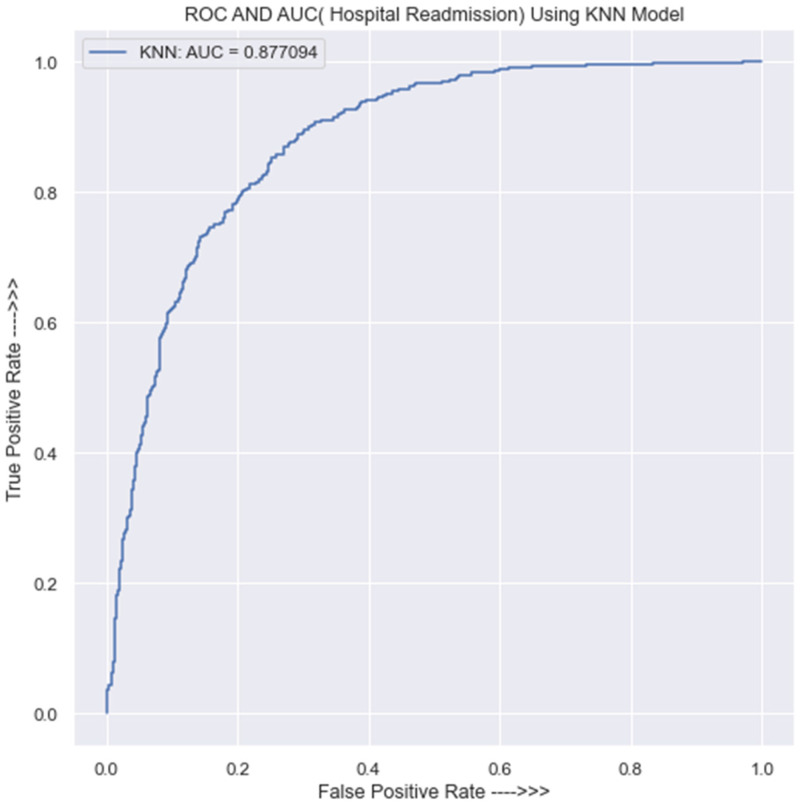
ROC curve for KNN classifier.

**Figure 8 jpm-13-01393-f008:**
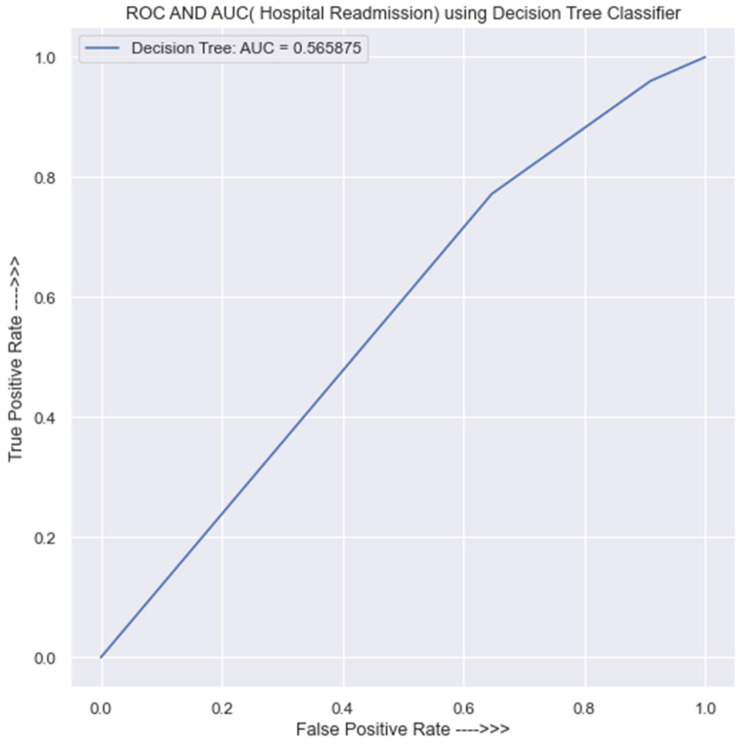
ROC curve for DT classifier.

**Figure 9 jpm-13-01393-f009:**
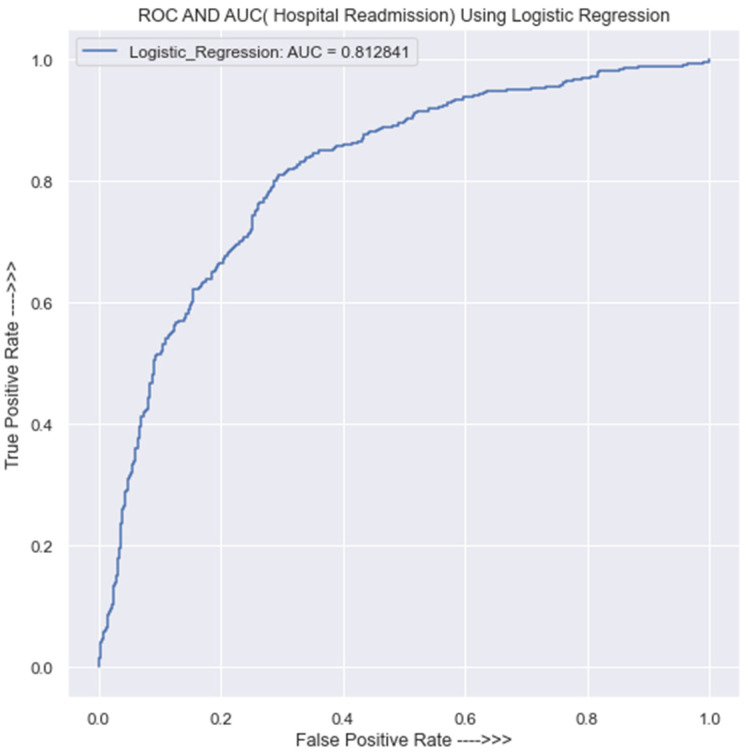
ROC curve for the LR classifier.

**Figure 10 jpm-13-01393-f010:**
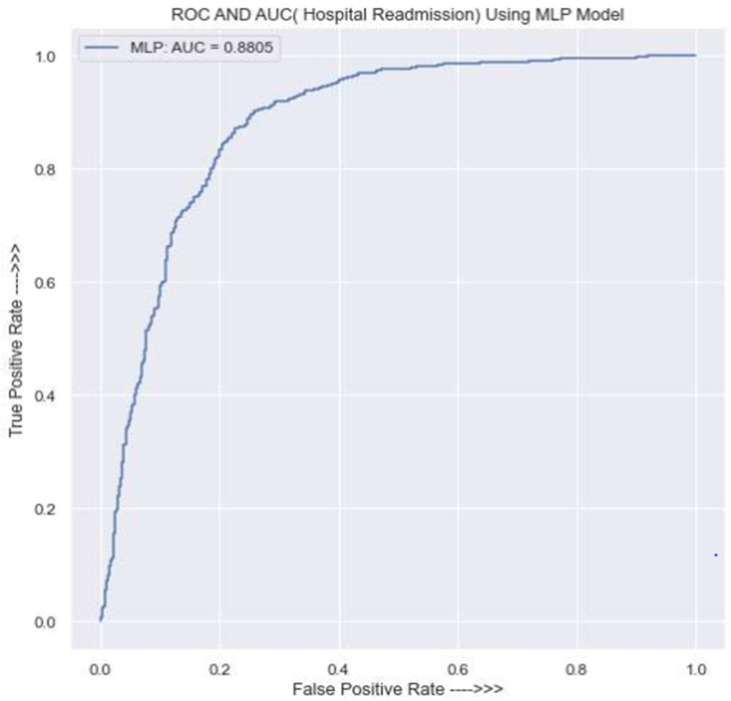
ROC for the MLP classifier.

**Figure 11 jpm-13-01393-f011:**
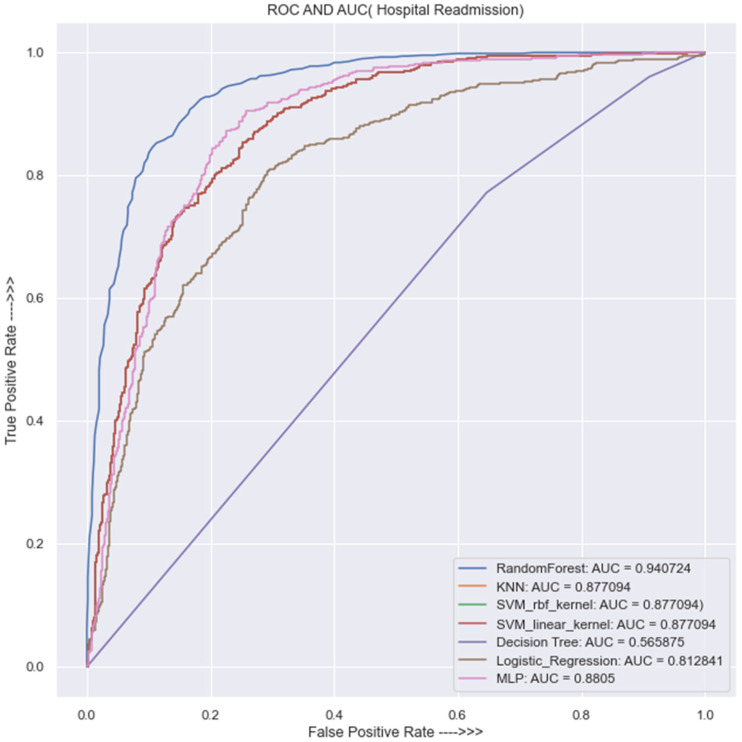
Receiver operating characteristics and area under the curve (ROC and AUC).

**Figure 12 jpm-13-01393-f012:**
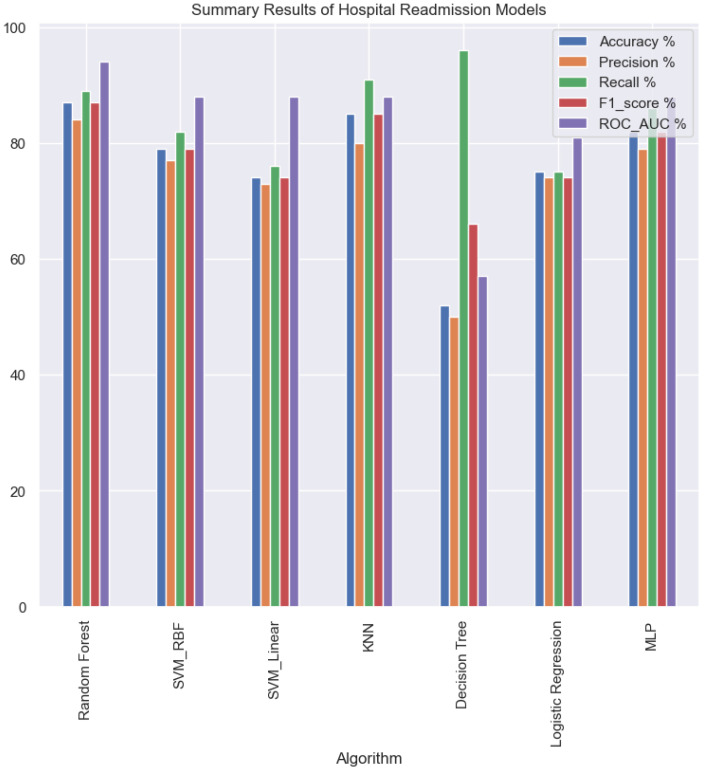
Bar plot comparison of model performance from Table 3.

**Table 1 jpm-13-01393-t001:** Sample dataset (5 rows × 75 columns).

S/N	Age	Sex	Alcohol	Risk_Decom_5	Hosp_Days	Max_Hra	Sbp_Maxrest	Dbp_Maxrest	…
0	6	0	0	0	3	128	94	62	
1	0	1	0	0	21	156	109	59	
2	1	0	0	0	53	170	113	88	
3	10	1	0	0	14	111	133	72	
4	15	0	0	0	35	142	120	90	

**Table 2 jpm-13-01393-t002:** Results of the target class before and after handling the class imbalance.

Class	Number of Classes before Balancing	Number of Classes after Balancing
**0 (no readmission)**	3469	3469
**1 (at least one readmission)**	614	3469
**TOTAL**	4083	6938

**Table 3 jpm-13-01393-t003:** Tabular summary of algorithm performance.

	Accuracy %	Precision %	Recall %	F1_Score %	ROC_AUC %
**Algorithm**					
**Random Forest**	87	84	89	87	94
**SVM_RBF**	79	77	82	79	88
**SVM_Linear**	74	73	76	74	88
**KNN**	85	80	91	85	88
**MLP**	82	79	86	82	88
**Logistic Regression**	75	74	75	74	81
**Decision Tree**	52	50	96	66	57

## Data Availability

The dataset used in this study is available from the corresponding author upon request.

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
