# Peer review of "Comparing Machine Learning Classifiers for Predicting Hospital Readmission of Heart Failure Patients in Rwanda"

_jpm, 2023, doi:10.3390/jpm13091393_

Round 1

Reviewer 1 Report

In this manuscript Theogene Rizinde et al compared different machine learning models to predict the heart failure readmission in Rwanda. Finally, they concluded that Random Forests (RF) performed best. Overall, the results were well presented. I totally agree with the claim that HF treatment between developing and developed countries is different. This paper provide a good strategies to predict the HF incidence in Rwanda.

Author Response

Having said that you totally agree with the claim that HF treatment between developing and developed countries is different and this paper provides good strategies to predict HF hospital readmission in  Rwanda, we thank you so much for the time you spent on this article and your valuable contribution.

Reviewer 2 Report

Rizinde and colleagues use machine learning to assess predictors of hospital readmission for HF in Rwanda.

While the concept is not novel, using the same to validate in a new population is shown here. I have the following suggestions to improve the presentation of the paper.

Introduction is too long and currently includes information from both methods as well as discussion currently. It should be focused; no longer than one page;  what is known and what is not in the specific topic and stating the objectives of the study to address the gaps.

Methods section thus, also need to expanded. Currently, results are discussing the methods.

Discussion of the results is found lacking and needs more emphasis on the clinical implications of the results. 

There needs to be more use of scientifc language for writing. for 1 example, "where the heart is no longer able  to pump blood through the body’s vessels at a rate sufficient to meet the body’s needs"

Round 2

Reviewer 2 Report

Comments has been addressed